# Harnessing the Native Extracellular Matrix for Periodontal Regeneration Using a Melt Electrowritten Biphasic Scaffold

**DOI:** 10.3390/jfb14090479

**Published:** 2023-09-19

**Authors:** Fanny Blaudez, Saso Ivanovski, Cedryck Vaquette

**Affiliations:** 1School of Dentistry, Centre for Oral Regeneration, Reconstruction and Rehabilitation (COR3), The University of Queensland, Herston, QLD 4006, Australia; fanny.blaudez@gmail.com (F.B.); s.ivanovki@uq.edu.au (S.I.); 2School of Dentistry and Oral Health, Griffith University, Southport, QLD 4222, Australia

**Keywords:** extracellular matrix, ECM decoration, polycaprolactone, melt electrowriting, additive manufacturing

## Abstract

Scaffolds have been used to promote periodontal regeneration by providing control over the spacio-temporal healing of the periodontium (cementum, periodontal ligament (PDL) and alveolar bone). This study proposes to enhance the biofunctionality of a biphasic scaffold for periodontal regeneration by means of cell-laid extracellular matrix (ECM) decoration. To this end, a melt electrowritten scaffold was cultured with human osteoblasts for the deposition of bone-specific ECM. In parallel, periodontal ligament cells were used to form a cell sheet, which was later combined with the bone ECM scaffold to form a biphasic PDL–bone construct. The resulting biphasic construct was decellularised to remove all cellular components while preserving the deposited matrix. Decellularisation efficacy was confirmed in vitro, before the regenerative performance of freshly decellularised constructs was compared to that of 3-months stored freeze-dried scaffolds in a rodent periodontal defect model. Four weeks post-surgery, microCT revealed similar bone formation in all groups. Histology showed higher amounts of newly formed cementum and periodontal attachment in the fresh and freeze-dried ECM functionalised scaffolds, although it did not reach statistical significance. This study demonstrated that the positive effect of ECM decoration was preserved after freeze-drying and storing the construct for 3 months, which has important implications for clinical translation.

## 1. Introduction

Periodontitis, a highly prevalent chronic inflammatory disease, results in extensive soft and hard tissue destruction of the tooth-supporting apparats known as the periodontium. The complex architecture of the periodontium makes periodontal regeneration a challenge. The most commonly used technique, guided tissue regeneration (GTR), utilises an occlusive membrane to selectively facilitate the re-population of the periodontal defect with cells derived from tissues that can promote periodontal regeneration (bone, periodontal ligament (PDL)) at the expense of those that cannot (gingival epithelium and connective tissue) [1]. Despite the demonstrated success of GTR in facilitating the regeneration of a small proportion of well-defined periodontal defects, the technique nevertheless remains unpredictable and cannot be applied to most periodontal defects [2,3,4]. The structural complexity of the periodontium suggests that the development of tissue-engineered solutions allowing the compartmentalised regeneration of both soft and hard tissues [5] may be a sound approach for achieving functional regeneration. In addition to achieving bone formation in the periodontal defect [6,7], it is essential to also facilitate the regeneration of the periodontal ligament and adjacent cementum, in order to meet the primary requirement of periodontal regeneration, which is the reconstitution of a functional periodontal attachment between bone and the tooth. Indeed, the periodontal ligament has been identified as the key requirement for periodontal regeneration, since it is composed of multipotent stem cells, involved in bone, periodontal ligament and cementum regeneration [8,9,10].

Additive manufacturing has been explored in the context of periodontal tissue engineering, allowing the fabrication of scaffolds with controlled structural properties that are capable of providing space maintenance and suitable for tissue infiltration and regeneration [11,12,13,14,15,16]. Many different printing techniques, polymers and structural parameters have been assessed for their performance in vivo. The development of multiphasic constructs [12,14,15,17,18,19,20] to allow the compartmentalised regeneration of soft and hard tissues has shown promising results. Indeed, the use of 3D-printed scaffolds for periodontal regeneration was also assessed in a clinical case, proposing a potential patient-specific approach [21]. Notably, most of the studies have utilised synthetic polymers (e.g., polycaprolactone (PCL) [12,16,18,19,20,21,22,23,24], polyglycolic acid (PGA) [15]), which, although biocompatible and biodegradable, lack biological activity and fail to achieve the structural and functional regeneration of the periodontium [14,15,16,18].

The biological enhancement of bioinert scaffolds, involving the addition of biological cues able to trigger a specific cellular response, has been extensively studied and described. This has been achieved by the addition of various bioactive molecules, such as growth factors [25,26,27,28,29,30] and RGD peptides [31]. Although highly potent, such enhancements do not address the native tissue’s architectural and molecular complexities and have been shown to achieve limited results. The use of native decellularised tissues may represent an interesting opportunity for tissue regeneration, as they contain all the required structural and biological properties and are stripped of any immunogenic cellular components [32]. Some approaches have investigated the use of decellularised whole tissue, either the entire mandibular bone (with teeth) [33] or teeth slices [34], but this is not particularly suitable for periodontal tissues due to their scarcity. The decellularisation of bone/ligament interfaces (entheses) is particularly challenging as each tissue will be differently impacted by the decellularisation protocol applied [35]. Thus, the emergence of scaffolds decorated with in vitro deposited ECM is of particular interest, as they possess both tailorable structures and a native-like biological environment. This may be achieved by way of coating 3D constructs with hydrogels made of solubilised native decellularised tissues [36,37]. However, native tissues require sterilisation to avoid pathogen transmission [38], and this will affect bioactivity. Overcoming this limitation, more recent strategies have explored the use of decellularised in vitro deposited ECM [39,40], which requires milder decellularisation treatments, allowing for a high preservation of biological cues. Cells can be directly cultured onto scaffolds and stimulated for enhanced ECM deposition, before being subsequently decellularised, resulting in a construct providing mechanical support and a natural ECM network [41,42]. This approach was pioneered by Pati et al. using a 3D-printed scaffold previously cultured with cells prior to decellularisation, and he coined the term “ornamented scaffold” [43]. This seminal paper demonstrated that ornamented scaffolds promoted bone formation in a rodent calvarial model. More recently, other research endeavours using 3D-printed scaffolds along with titanium implants revealed that cell-laid extracellular matrix decoration resulted in beneficial outcomes for angiogenesis [44] and osteogenesis [45,46]. 

The use of periodontal ligament cell sheets (PDLcs’s) has been identified as an efficient way to facilitate cell delivery while maintaining an intact mature ECM [12,24,47,48,49,50,51,52,53]. Previous work by our group has explored the decellularisation of PDLcs’s in combination with a PCL melt electrowritten scaffold [41,42,54,55]. We demonstrated successful periodontal ligament attachment as well as bone formation within the PCL fibres, promoting overall construct integration and periodontal regeneration [41]. In the present study, a previously optimised decellularised scaffold for bone formation [56] is combined with a decellularised periodontal fibroblast cell sheet to develop a biphasic scaffold promoting compartmentalised regeneration for the periodontium. This proof-of-concept study combines a periodontal ligament cell sheet with a bone-like ECM-decorated scaffold before decellularisation and may ultimately generate an off-the-shelf biphasic scaffold capable of assisting periodontal regeneration. In addition, the medium-term storage effect on the regenerative performance of the freeze-dried biphasic decellularised construct is assessed in a surgically created periodontal rodent model. 

## 2. Materials and Methods

### 2.1. Cell Isolation and Culture

Primary human osteoblasts (hOBs) and periodontal ligament (hPDL) cells were explanted from redundant tissues collected from a patient undergoing dental surgery according to an established protocol [57,58]. Briefly, alveolar bone fragments removed during tooth extraction were isolated and chopped into small segments, covered with a complete medium (DMEM high glucose, invitrogen) supplemented with 10% Fetal Bovine Serum (FBS) and 4% Penicillin-Streptomycin (reduced to 1% after 24 h) and incubated at 37 °C with 5% CO2. Similarly, periodontal ligament tissue was obtained from the middle third of tooth roots, finely chopped and expanded in a complete medium. Cells at about 80% confluence and ranging from passage 4 to passage 6 were utilised for all in vitro and in vivo studies. 

### 2.2. Melt Electrowritten Poly(ε-Caprolactone) Scaffold

Melt electrowritten (MEW) scaffolds were fabricated via a direct writing approach previously described [59], whereby a programmable x-y stage was used to collect the fibres. Briefly, medical-grade poly(ε-caprolactone) (MPCL) (PC12, Corbion) pellets were placed into a 2 mL syringe and heated at 80 °C in an oven to melt the polymer. The molten polymer was electrowritten through a blunt 21 G needle, a pressure of 1.6 bar and a voltage of 9 kV with a spinneret collector distance of 8 mm. The temperatures of the first heater (placed near the syringe) and the second heater (placed near the needle) were set to 75 °C and 85 °C, respectively. The translational speed of the collector was set at 850 mm/min in order to obtain straight fibres, and a rectilinear pattern was utilised for fabricating a scaffold composed of alternating series of layers oriented at 0 and 90° with a 250 µm fibre interdistance. The resulting membranes (1.5 mm thick) were sectioned with a scalpel blade into 5 mm squares. The melt electrowritten scaffolds were etched with 2 M NaOH for 60 min at 37 °C in order to increase hydrophilicity and roughness, facilitating fluid infiltration and subsequent cell attachment [60]. The scaffolds were thereafter rinsed several times in distilled water to remove any excess NaOH. The scaffolds were sterilised by immersion in 80% ethanol for 60 min and then dried overnight in a biosafety cabinet prior to a 30 min UV exposure. 

### 2.3. Fabrication of the Bone Compartment

The scaffolds were immersed in FBS prior to cell seeding to improve cell colonisation and maturation, as previously described [61]. The scaffolds were then incubated for 2 h at 37 °C in FBS before the removal of excess liquid. Cells (hOBs) were harvested via incubation for 5 min in 1% trypsin and then resuspension in a complete medium at a concentration of 2 × 106 cells/mL, and the constructs were seeded with 100,000 cells in 50 µL of the complete medium. The cells were then allowed to attach for 2 h and the scaffolds were transferred to 48-well plates containing 500 µL of a complete medium. The cells were cultured for 3 days in a basal medium before osteogenic induction was initiated by supplementing the medium with 50 µg/mL of Ascorbic Acid (AA), 0.1 µM of Dexamethasone and 10 mM β-Glycerophosphate (osteogenic medium). The osteogenic medium was changed every 3 days, and the cells were cultured for 1 week as per our optimised protocol [56].

### 2.4. Periodontal Ligament Cell Sheet Formation and Harvesting

The hPDL cell sheets (hPDLcs’s) were prepared following a previously optimised protocol [54]. hPDL cells were seeded in 24-well plates at a seeding density of 5 × 10^4^ cells/well in 750 µL of a basal medium and cultured for 3 days in the basal medium before being supplemented with ascorbic acid (100 μg/mL; Sigma-Aldrich, Castle Hill, NSW, Australia). The medium was changed every other day. After 21 days of culture, the cells had deposited a large amount of ECM, forming a cell sheet that could be manually handled and harvested. In order to combine the bone and PDL compartments, the 1-week-old cellularised bone scaffold was placed in the centre of the well containing the hPDLcs, and the borders of the cell sheet were gently detached from the base of the well and folded over the edges of the bone–PCL construct using sterile tweezers as previously reported [12,20]. The resultant biphasic construct was placed in an expansion medium for 24 h with the cell sheets facing upward, in order to allow cell sheet adhesion onto the cellularised 1-week-old bone scaffold.

### 2.5. Decellularisation Protocol

The following protocol was previously optimised and assessed for cell removal efficiency as well as matrix preservation [54]. Every step of the decellularisation process was carried out under sterile conditions, at 37 °C on an orbital shaker (100 rpm) for 1 h, and each treatment was followed by three washes in phosphate-buffered solution (PBS). Briefly, the biphasic scaffolds were first treated with 400 µL of 20 mM NH4OH solution with 0.5% Triton X-100. Then, remnant DNA was removed by using DNase I solution (100 U/mL, Invitrogen, Melbourne, VIC, Australia) in PBS. Finally, the scaffolds were incubated in milliQ water in order to remove the remaining cell fragments and chemicals to prevent any cytotoxicity or immunogenicity of the constructs.

### 2.6. In Vitro Characterisation

#### 2.6.1. Confocal Microscopy

Biphasic scaffolds, both before and after decellularization (*n* = 3), were rinsed in PBS then fixed for 30 min in 4% paraformaldehyde (PFA) in PBS. The scaffolds were then incubated for 60 min in blocking buffer (1% bovine serum albumin; 10% goat serum; 0.05% tween-20 solution in PBS) at room temperature (RT) and subsequently incubated for 1 h at RT with collagen I mouse primary antibody (1:200, ab6308, Abcam, Melbourne, VIC, Australia) diluted in blocking buffer or collagen VI rabbit primary antibody (1:200, ab182744, Abcam) diluted in blocking buffer. Samples were then washed 3 times with PBS and incubated for 60 min with their respective secondary antibodies; goat anti-mouse IgG Alexa Fluor^®^ 488-conjugated secondary antibody (1:500, ab150113, Abcam) or donkey anti-rabbit IgG Alexa Fluor^®^ 647-conjugated secondary antibody (1:500, ab150075, Abcam) diluted in blocking buffer. All samples were then counterstained with 5 μg/mL 4,6-diamino-2-phenylindole (DAPI, Life Technologies, NY, USA) for 10 min at RT to visualise remnant DNA. The scaffolds were then rinsed in PBS and imaged using a confocal microscope (Nikon, Eclipse-Ti, Melville, NY, USA) at an excitation/emission wavelength of 405/417-477 for DAPI, 488/500-550 nm for collagen I (green) and 561/570-1000 nm for collagen VI (red).

#### 2.6.2. Scanning Electron Microscopy

Prior to and after decellularisation, the biphasic scaffolds were rinsed in PBS and fixed in 2.5% Glutaraldehyde in Cacodylate buffer overnight. The scaffolds were then serially dehydrated using ethanol solutions of increasing concentrations (50%, 70%, 80%, 90%, 95% and 100%). The scaffolds (*n* = 3) were then mounted onto SEM stubs and carbon coated in vacuum using a sputter coater (Balzers SCD 004, Wiesbaden-Nordenstadt, Germany). Scanning electron microscopy (SEM) was carried out using a JSM F-7001 microscope (JEOL Ltd., Tokyo, Japan), operating at 15 kV and 10 mm of working distance. 

### 2.7. In Vivo Study

#### 2.7.1. Preparation of Implants 

Four groups were evaluated in vivo: (a) an untreated defect (empty), (b) a pristine scaffold (that is, without ECM cell-laid decoration) (scaffold), and (c) a decellularised biphasic construct either fresh (cultured for 1 week prior to decullarisation, which was performed on the day of the implantation) (1W-decel) or (d) freeze-dried (FD). The freeze-dried constructs were prepared 3 months before the surgeries and stored in a desiccator at room temperature. The freeze-drying process was performed as follows: after decellularisation, excess milliQ water was removed from the decellularised scaffolds, which were then placed overnight at −80 °C and then freeze-dried in sterile conditions (0.005 mbar, −80 °C, FreeZone 2.5 Plus, Labconco) for 12 h. The constructs were rehydrated in PBS for 5 min before implantation.

Freshly decellularised scaffolds were decellularised on the day of surgery and placed in PBS until implantation.

#### 2.7.2. Rat Periodontal Defect

The study protocol was approved by the Animal Ethics Committee of the University of Queensland (DENT/274/19). Ten–twelve-week-old Sprague Dawley rats (Animal Resources Centre, Canning Vale, WA, Australia) were carefully placed in an induction chamber and anaesthetised via inhalation, using a mixture of oxygen and Isoflurane (AttaneTM, Bomac Animal Health Pty Ltd., Hornsby, NSW, Australia). The periodontal defect model was adopted from a previous study [62]. Briefly, a full thickness incision was made along the skin of the inferior border of the mandible. The masseter muscle and the periosteum were reflected, followed by the preparation of alveolar bone defects to the dimensions of 2 mm height × 3 mm width and 1 mm in depth [62] using a small (1 mm) round bur and copious water irrigation. Mandibular defects were then randomly assigned to either remain untreated (empty) or receive a pristine scaffold or the decellularised biphasic construct either fresh or freeze-dried. Upon implanting the FD construct, we did not observe any obvious morphological differences from the fresh scaffolds once it was re-hydrated. ECM-decorated constructs were placed so the PDL sheet was facing the exposed roots. A resorbable bilayer membrane (Bio-Gide^®^, Geistlich, Princeton, NJ, USA) was then used to cover the scaffold in order to exclude the overlying connective tissue from the defect, according to the principles of “guided tissue regeneration” (GTR).

The animals were sacrificed at 4 weeks post-surgery, and the mandible samples (n = 5 for each group) were collected and fixed in 4% paraformaldehyde (PFA) solution overnight at room temperature and then rinsed with PBS.

#### 2.7.3. High-Resolution Microcomputed Tomography (microCT)

High-resolution microCT imaging was performed using a Skyscan 1272 (Bruker, Billerica, MA, USA) to scan the rat mandibles. Rat mandible specimens (n = 5), stored in PBS, were wrapped in moist tissue paper and transferred into 5 mL cylindrical plastic tubes for imaging. The scanning parameters were a 90 kV X-ray voltage, 111 μA current, 1800 ms exposure time, 21 μm isotropic voxel size, 0.6° rotation step (360° imaging), 3 frame averaging, 4 × 4 binning, and 0.5 mm Al filter. The datasets were reconstructed with NRecon (version 1.7.3.1; Bruker, Billerica, MA, USA) and InstaRecon (version 2.0.4.2; University of Illinois, Champaign, IL, USA) software using a cone beam reconstruction (Feldkamp) algorithm with the following corrections applied: ring artefact reduction, smoothing, beam hardening and post-alignment. CT analysis was performed using CTan Software version 1.19.11.1 (Bruker, Billerica, MA, USA) and 3D visualisations of were mandibles generated using CTVox Software (version 2.6; Bruker, Billerica, MA, USA). 

### 2.8. Histological Analysis

The mandible samples (*n* = 5) were decalcified in 10% (*w*/*v*) EDTA at 37 °C and a pH of 7.4 for one month in a multifunctional microwave tissue processor (Milestone Medical, Kalamazoo, MI, USA) with weekly changes of the solution prior to embedding in paraffin. Consecutive serial sections (5 µm) were subsequently deparaffinised and stained with Masson’s Trichome (TRI Histology facility with Tissue-Tek automated stainer and coverslipper). The stained slides were scanned using a 40× objective lens on a virtual slide microscope (VS120; Olympus, Tokyo, Japan) for subsequent morphological analysis.

### 2.9. Quantification of New Attachment 

Newly formed cementum and new periodontal ligament attachment on the surgically denuded roots was quantified as previously described [62]. Cementum formation was determined as the percentage of the root surface that was covered by newly formed cementum. For periodontal attachment measurement, the length of the tooth surface with inserting periodontal fibres oriented with an angle above 60° to the root surfaces (measured as the angle between the long axis of the fibres and the root surface) was divided by the total length of the denuded root surface. The borders of the root defect could be easily differentiated from the intact native periodontium. The OlyViIA^®^ 3.3 software (Olympus) was used to visualise the slides and perform the measurements.

### 2.10. Statistical Analysis

Statistical analysis was performed using GraphPad Prism (version 8.2.1). Since the dataset consisted of multiple groups and/or time points, multiple comparisons were performed using a one-way ANOVA followed by a Tukey post hoc test. 

## 3. Results

### 3.1. In Vitro Characterisation

The decellularised bone compartment and PDLcs have been separately optimised and extensively characterised in previous studies from our group [42,54]. The following experiments aim to verify the integrity of the ECM and the effective removal of its cellular components after both compartments were assembled.

The SEM images confirmed that the attachment of the cell sheet to the cellularised bone compartment was maintained during the decellularisation process (Figure 1). Cells could be identified in the fresh scaffolds, embedded in their collagenous matrix, while they were absent on the decellularised constructs. The structural integrity of the ECM was observed to be preserved after decellularisation (Figure 1).

The staining of DNA, collagen I and collagen VI using immunofluorescent imaging confirmed the aforementioned observations (Figure 2). In both compartments, bone and cell sheet, efficient cellular removal was confirmed by the absence of a DAPI signal. The overall structures of the collagen I and the collagen VI networks were similar in the cellularised and decellularised scaffolds, confirming an overall good preservation of the ECM.

### 3.2. In Vivo Study

#### 3.2.1. Bone Formation 

All groups performed similarly in terms of new bone formation (Figure 3) with a defect bone fill ranging from 10 to 20%. 

#### 3.2.2. Histological Analysis

Masson’s trichrome staining allowed the identification of the anatomical structure of the regenerated periodontium, including newly formed bone, cementum and periodontal ligament (Figure 4A). As shown in Figure 4A, there was some significant natural healing in the empty defect, with the presence of newly formed bone. Similarly, the scaffold group demonstrated bone formation and some cementum-like issue was observed on the previously exposed root surface. The scaffold displayed good space maintenance and the PCL scaffold was fully integrated in the surrounding tissues with the formation of cementum and PDL-like tissues (Figure 4A). Both the 1W-decel and the freeze-dried groups presented similar results and also presented full periodontal regeneration. The percentage of newly formed cementum was measured and demonstrated that all groups displayed a high percentage of new cementum formation (empty: 60 ± 31%, scaffold: 46 ± 12%, 1W-decel: 82 ± 18%, FD: 76 ± 21%). While there was no statistical significance between the groups, the decellularised groups constantly displayed higher average cementum formation. The periodontal regeneration followed a similar trend; indeed, the percentage of periodontal attachment ranged from 20% to 50% (empty: 30 ± 16%, scaffold: 22 ± 13%, 1W-decel: 49 ± 19%, FD: 43 ± 18%). Here again, both decellularised groups displayed higher percentages of periodontal attachment when compared to the other groups, although it did not reach statistical difference.

Values are presented as individual data points, superimposed by the mean (horizontal line) and error bars showing plus and minus one standard deviation (SD) (as represented by the whiskers).

Interestingly, we observed a better preservation of the space between the newly formed bone and cementum when a scaffold was implanted and more specifically when decellularised ECM was utilised (Figure 5). This resulted in a smaller periodontal gap in the empty group as shown in Figure 5. This increased wound stability seemed to have a significant impact on the formation and frequency of bone formation in direct contact with the dentine. Indeed, the empty group displayed a higher occurrence of ankylosis (Table 1) as shown in Table 1, while it was not detected in any of the decellularised specimens (freshly decellularised or freeze-dried). This is an undesirable outcome that would compromise the longevity of a tooth.

## 4. Discussion

This study assessed the safety and performance of the decellularised scaffold performance for periodontal regeneration. The application of an established decellularisation protocol to the biphasic construct was successful in preserving the extracellular matrix of both compartments. As previously demonstrated, the utilised protocol ensures optimum retention of biological cues due to a mild yet effective chemical treatment [42,56]. The primary finding of the present study revealed that the implantation of decellularised scaffolds via mild chemical treatments did not result in an increased inflammatory response when compared to the controls. The utilisation of well-preserved decellularised ECM is a crucial component enabling tissue regeneration. Indeed, the bioactivity is mediated via the release of biological factors embedded in the ECM [63,64,65]; therefore, mild but efficient decellularisation strategies allow for an enhanced preservation of these molecules.

In the present study, a biphasic scaffold strategy was applied, consistent with the previous literature [12,15,16,18,19,20,22,23,66]. Indeed, biphasic scaffolds are envisioned to guide the spatio-temporal regenerative processes leading to periodontal regeneration [22,23,67]. In this strategy, two different cell types were utilised for developing a decellularised scaffold with the ability to provide tissue specific cues to trigger compartmentalised tissue regeneration. In order to enhance the cohesion between both compartments, the cell sheet was cultured for 24 h prior to decellularisation. This ensured that cell attachment occurred and resulted in an increased stability at the bone–PDL compartment interface. While our previous study reported poor bone regeneration when a thick decellularised ECM was present, the cell sheet was efficiently decellularised, consistent with our previous reports [41,54]. Indeed, the decellularisation of the biphasic scaffold results in similar outcomes when decellularising the PDLcs alone either in static or perfusion conditions [54].

Although in vitro characterisation displayed a good preservation of the ECM and an absence of cell nuclei, the decellularised scaffolds did not induce significantly more bone formation compared to the controls, which was inconsistent with the findings of previous studies [43,56,62]. This discrepancy may be due to the nature (location, size and type) of the periodontal defect, which is surrounded by resident bone. Hence, the surgically created periodontal window defect is more prone to spontaneous natural healing. It should also be noted that the periodontal defect was utilised primarily to assess the effect of the biphasic construct on periodontal regeneration, rather than bone. Indeed, the dimensions of the rat periodontium do not allow for the creation of a “critical size” bone defect per se, although the defect is of “critical size” in the context of periodontal regeneration, in that its full periodontal regeneration does not occur in the untreated defect, at least not in the timeframe relevant to this study [68]. The assessment of the decellularised scaffold in a larger animal model may thus provide further insight onto the bone regenerative capacity of the construct.

Although decellularised ECM constructs did not induce higher levels of bone formation, they displayed promising potential for the functional regeneration of the periodontium. Although the number of replicates did not allow a significant difference to be reached, both fresh and freeze-dried constructs resulted in higher mean newly formed cementum, allowing for higher periodontal ligament attachment. This result, consistent with previous characterisations [41], is likely to be due to the presence of the decellularised cell sheet, which potentially provided a suitable micro-environment for cementogenesis to occur, subsequently leading to periodontal attachment. The regeneration of the alveolar bone and cementum is required to allow periodontal attachment and hence to achieve the functional and structural regeneration of the periodontium [68,69]. However, the undesirable presence of bone directly against the dentine of the surgically denuded tooth root, known as ankylosis, was observed in some of the controls (empty and pristine scaffolds) and is a limiting factor of periodontal regeneration as it prevents the formation of a functional periodontal ligament [70,71]. The presence of ankylosis results in osteoclast recruitment, which can later lead to root resorption [70], and hence the absence of ankylosis in samples containing ECM is a very promising feature for achieving periodontal regeneration. In this context, the presence of the thick decellularised cell sheet is likely to have been beneficial for the maintenance of the periodontal ligament space, which is essential for the formation of attachment leading to periodontal regeneration. 

As previously highlighted in other reports, the utilisation of cell therapy for periodontal regeneration, while potentially efficient [50,72], may be difficult to implement and expensive in the clinical setting [20,73]. Therefore, the utilisation of decellularised scaffolds can circumvent some of those limitations by enabling larger batch production and enhanced quality control. However, the major drawback of the decorated scaffold relates to logistical hurdles of having an appropriate facility in the vicinity of the point of care for cell cultures and the processing of the constructs. Indeed, in their hydrated and freshly decellularised form, these constructs need to be implanted immediately post-manufacturing to prevent biological cue denaturation, thus drastically limiting their shelf-life. Therefore, a common biological sample preservation technology (freeze-drying) was trialled in the present study. The freeze-drying of biological constructs has been widely used to generate anhydrous implants, which increase ECM preservation and facilitate storage [74] while maintaining the graft biochemical and mechanical properties [33,75]. Importantly, the present study demonstrated that there was no or negligible denaturation of the biologically active components as both freshly decellularised and freeze-dried constructs performed equivalently in vivo in regards to bone formation, cementum formation and periodontal ligament attachment. This comparison demonstrated that the freeze-drying and storage of the decellularised scaffolds did not induce any negative outcomes or significant loss of bioactivity. This enables an off-the-shelf approach that alleviates many of the impediments related to the storage and shipping of these potential products. Thus, it is a promising outcome for the clinical translation of ECM-decorated engineered constructs. Lastly, the utilisation of melt electrospun scaffolds is a promising strategy for periodontal regeneration due to its adaptability, ensuring close proximity to the root surface and acting as a substrate with excellent handling properties for the delivery of cell-laid ECM.

## 5. Conclusions

This study utilised the concept of scaffold decoration for periodontal regeneration, which was achieved by separately culturing two types of cells prior to decellularisation to enable a tissue-specific and compartmentalised regenerative response. The decellularisation of the biphasic scaffold successfully generated an acellular construct with the preservation of the in vitro deposited ECM. All groups were capable of supporting bone formation and periodontal regeneration was similar across the groups, while the decellularised groups displayed higher periodontal regeneration. In addition, the presence of a decellularised cell sheet was beneficial for the maintenance of the PDL space, preventing ankylosis, which occurred in some of the empty and scaffold-only groups. This suggested that the biphasic decellularised scaffold facilitated periodontal regeneration in a coordinated compartmentalised manner. The key finding of this study is that the further freeze-drying of the biphasic scaffold did not alter the biological performance of the bioactive molecules deposited by cells on the scaffold.

## Figures and Tables

**Figure 1 jfb-14-00479-f001:**
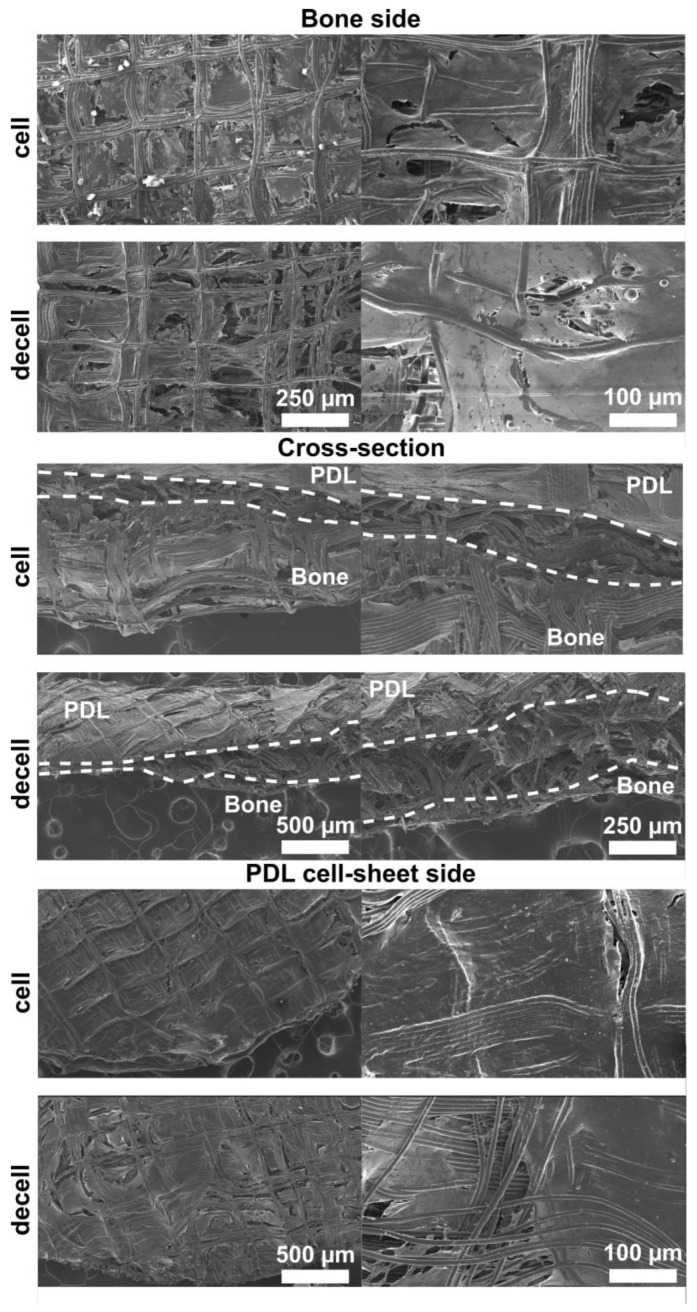
Scanning electron microscopy images of the cellularised (cell) and decellularised (decell) biphasic constructs. Images were taken at 50, 100 and 250 magnifications to characterise the surface of the bone compartment, the PDL cell sheet and the cross-section of the construct. The PDL and bone compartment of the cross-sectional view have been delimited by the dotted line and, respectively, labelled with “PDL” and “Bone”.

**Figure 2 jfb-14-00479-f002:**
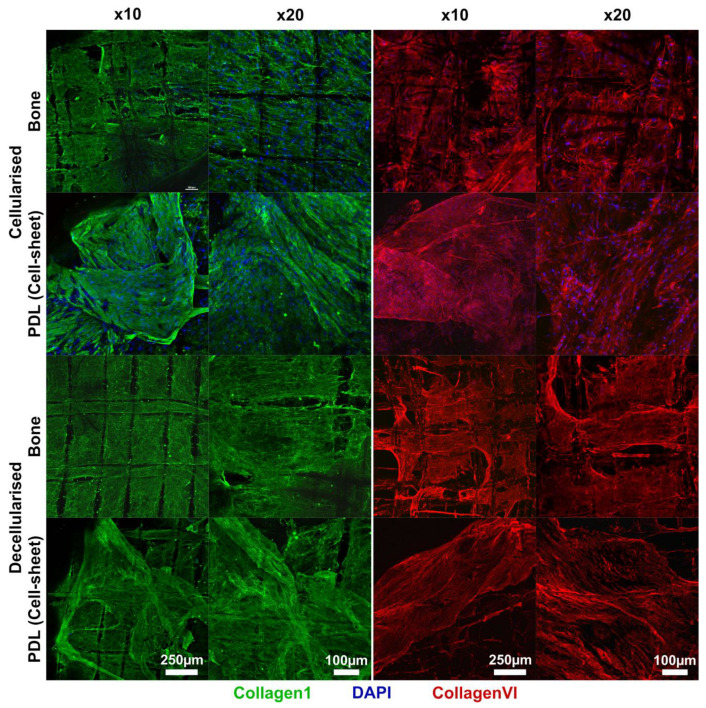
Confocal image with ×10 and ×20 magnification of the biphasic scaffold, displaying views from the bone side and the periodontal ligament side. Samples were stained for cellular DNA in blue (DAPI), collagen VI in red and the main ECM component, collagen I, in green.

**Figure 3 jfb-14-00479-f003:**
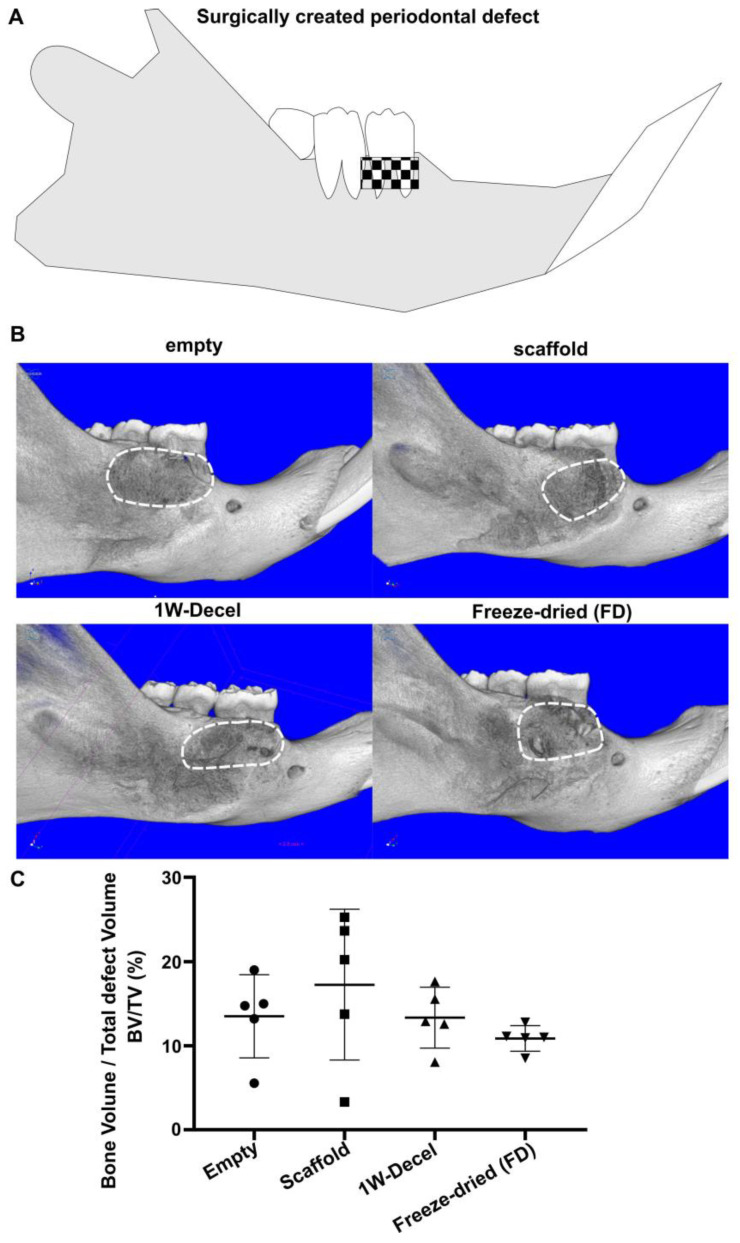
(**A**) Schematic of the periodontal defect (**B**) MicroCT images of the different groups along with (**C**) the quantification of the newly formed bone in the defect area (as indicated by the white dash line). Values are presented as individual data points, superimposed by the mean (horizontal line) and the whiskers representing one standard deviation (SD).

**Figure 4 jfb-14-00479-f004:**
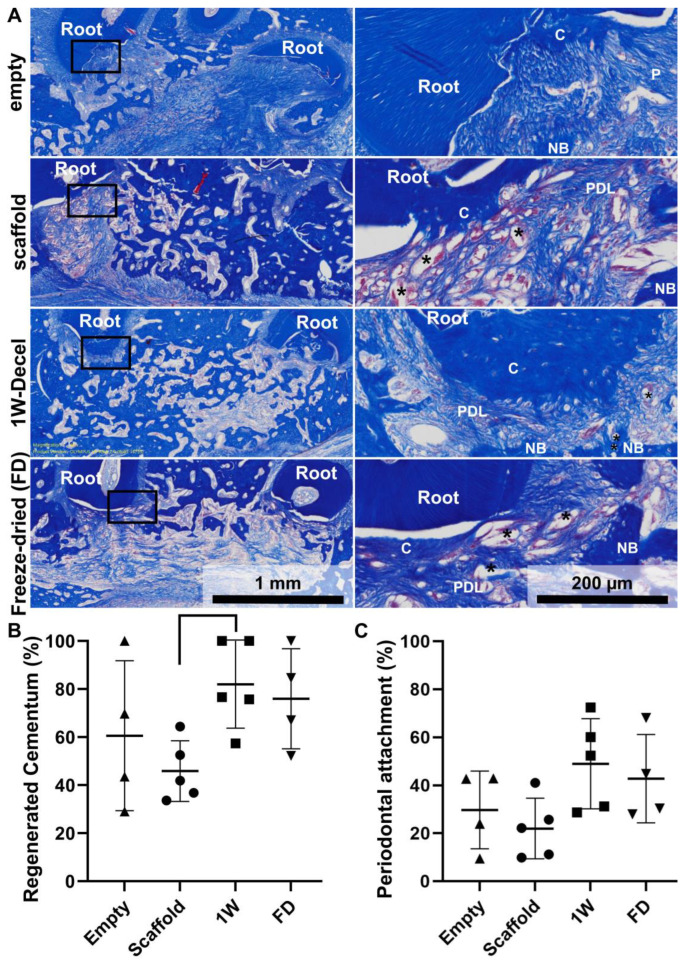
(**A**) Masson’s trichrome staining of the different groups at magnification ×2 (left panel), where the areas designated by the box have been magnified ×10 (right panel). Newly formed cementum is indicated by “C”, bone by “B”, PDL by “PDL” and the PCL fibres by “*”. (**B**) The percentage of newly formed cementum on the surgically denuded roots was quantified, along with (**C**) the percentage of regenerated periodontal attachment.

**Figure 5 jfb-14-00479-f005:**
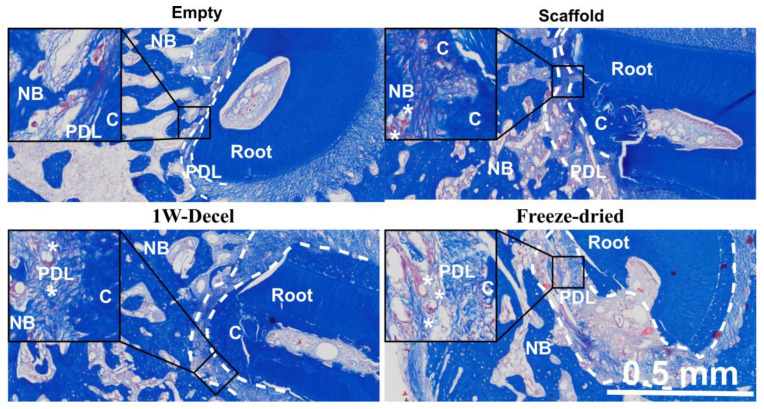
Masson’s trichrome staining of the different groups showing periodontal regeneration in the defect area at magnification ×4, where the areas designated by the box have been magnified ×20. Newly formed cementum is indicated by “C”, new bone by “NB”, periodontal ligament by “PDL” and the PCL fibres by “*”. The white dashed lines represent the periodontal gap.

**Table 1 jfb-14-00479-t001:** The frequencies of bone forming in direct contact with the cementum or dentin were reported.

Group	Frequency of Ankylosis
Empty	2/5
Scaffold	1/5
1W-decel	0/5
FD	0/5

## Data Availability

Data can be made available upon request.

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
