# Peer review of "Harnessing the Native Extracellular Matrix for Periodontal Regeneration Using a Melt Electrowritten Biphasic Scaffold"

_jfb, 2023, doi:10.3390/jfb14090479_

Round 1
Reviewer 1 Report
1. In the “Introduction” part, the authors introduced the rationale of “decellularised in vitro deposited ECMs”, but the background of this kind of scaffold was not sufficiently provided.
2. In line 88-89, please provide proper literatures to support the statement “The use of periodontal ligament cell sheets (PDLcs) has been identified as an efficient way to facilitate cell delivery while maintained an intact mature ECM.”
3. Please clarify that the “1 week old bone-scaffold” used in combining the bone and PDL compartments (section 2.4) was the acellular scaffolds or the scaffold-cell constructs. For the later case, it should not be referred as “1 week old bone-scaffold” since a scaffold was generally regarded as a structure of material without cells.
4. In Fig. 1, the cross-section showed the bi-layer structure, but the bone and the PDL compartments seemed very similar. Please give out a reasonable explain.
5. It would be better to perform a quantitative detection of DNA after the decellularization.
6. It is better to provide an intuitionistic diagram or image of the defect and the implanted scaffold to illustrate the surgery site and the scaffold location.
7. Please mark the defect area in Fig. 3A. Also, the amplified cross-section of the defect area should be showed. The present Fig. 3A displayed very limited information, and did not distinguish different groups.
8. It was not clear what was the pristine scaffold. In addition, in the text, it was described that the “Freshly decellularised scaffolds were decellularised on the day of surgery”, but in the figures, there were groups of “1W-Decel”. Please make sure which one was correct.
9. Did the mandible samples for Fig. 4 and 5 include the teeth itself?Please explain the sampling location and area. In these images, root (dentin), cementum, PDL, and bone were not clearly identified.
The language of this manuscript is generally acceptable. But some minor typo- and misuse were detected, and should be corrected.
Reviewer 2 Report
1. It is recommended to provide an appropriate explanation when mentioning "1W-Decel" for the first time in the article to help readers understand.
2. Figure 3A is missing the initial image of a post-operative periodontal defect, and the abbreviation “BV/TV (%)” on the Y-axis in Figure 3B is not clearly explained, which can lead to difficulties in comprehending the information.
3. In Figure 4B, the Y-axis label “% regenerated cementum” should be modified to “Regenerated cementum (%)” to maintain consistency with Figure 3B.
4. It is recommended to include a scale bar in Figure 5A of the tissue analysis section to achieve consistency with Figure 4A.
5. It is suggested to label Figure 5B as Table 1. 4. Background descriptions can be strengthened by citing 10.1016/j.carbpol.2020.116585; 10.1016/j.cej.2023.141852.
6. The format of the data in Page 10, line 307 “Empty: 60%±31, Scaffold:.46%±12, 1W-Decel:82%±18, FD: 76%±21” is inconsistent. It would be better to standardize the format.
7. The abbreviation format of “PDLCS” (Page 13, line 364) is inconsistent with the earlier abbreviation “PDLcs” used in the text. It is recommended to make the necessary modification to maintain consistency in the abbreviation throughout the entire document
8. The introduction of the article mentions the concept of providing mechanical support through the biphasic scaffold. However, the experimental study did not include any testing of the mechanical properties of the scaffold, which may be considered insufficient.
Reviewer 3 Report
This study demonstrated that the positive effect of ECM decoration was preserved after freeze-drying and storing the construct for 3 months, which has important implications for clinical translation.
However, the scale bar of the SEM images is ambiguous and needs to be rechecked.
Moreover, the authors didn't clearly demonstrate the terms cell and decell in Figure 1.
Confocal images in Figure 2 also need to be rechecked in terms of scale bars.
Why the authors didn't show the morphology of the scaffolds; freeze-drying affects the morphology of the scaffolds. It would be interesting to provide the morphology (SEM or CLSM) images of the as-prepared and freeze-dried scaffolds.
